# Evaluation of Progressive Architectural Distortion in Idiopathic Pulmonary Fibrosis Using Deformable Registration of Sequential CT Images

**DOI:** 10.3390/diagnostics14151650

**Published:** 2024-07-31

**Authors:** Naofumi Yasuda, Tae Iwasawa, Tomohisa Baba, Toshihiro Misumi, Shihyao Cheng, Shingo Kato, Daisuke Utsunomiya, Takashi Ogura

**Affiliations:** 1Department of Radiology, Kanagawa Cardiovascular and Respiratory Center, 6-16-1 Tomioka-Higashi, Kanazawa-ku, Yokohama 236-0051, Kanagawa, Japan; yasuda.nao.mz@yokohama-cu.ac.jp; 2Department of Diagnostic Radiology, Yokohama City University Graduate School of Medicine, 3-9, Fukuura, Kanazawa-ku, Yokohama 236-0004, Kanagawa, Japan; a9613097@gmail.com (S.C.); sk513@yokohama-cu.ac.jp (S.K.); d_utsuno@yokohama-cu.ac.jp (D.U.); 3Department of Respiratory Medicine, Kanagawa Cardiovascular and Respiratory Center, 6-16-1 Tomioka-Higashi, Kanazawa-ku, Yokohama 236-0051, Kanagawa, Japan; baba.19049@kanagawa-pho.jp (T.B.); takaoguogu@gmail.com (T.O.); 4Department of Biostatistics, Yokohama City University Graduate School of Medicine, 3-9, Fukuura, Kanazawa-ku, Yokohama 236-0004, Kanagawa, Japan; misumit@yokohama-cu.ac.jp

**Keywords:** idiopathic pulmonary fibrosis, computed tomography, deformable image registration, three-dimensional average displacement, progressive pulmonary fibrosis

## Abstract

Background: Monitoring the progression of idiopathic pulmonary fibrosis (IPF) using CT primarily focuses on assessing the extent of fibrotic lesions, without considering the distortion of lung architecture. Objectives: To evaluate three-dimensional average displacement (3D-AD) quantification of lung structures using deformable registration of serial CT images as a parameter of local lung architectural distortion and predictor of IPF prognosis. Materials and Methods: Patients with IPF evaluated between January 2016 and March 2017 who had undergone CT at least twice were retrospectively included (*n* = 114). The 3D-AD was obtained by deformable registration of baseline and follow-up CT images. A computer-aided quantification software measured the fibrotic lesion volume. Cox regression analysis evaluated these variables to predict mortality. Results: The 3D-AD and the fibrotic lesion volume change were significantly larger in the subpleural lung region (5.2 mm (interquartile range (IQR): 3.6–7.1 mm) and 0.70% (IQR: 0.22–1.60%), respectively) than those in the inner region (4.7 mm (IQR: 3.0–6.4 mm) and 0.21% (IQR: 0.004–1.12%), respectively). Multivariable logistic analysis revealed that subpleural region 3D-AD and fibrotic lesion volume change were independent predictors of mortality (hazard ratio: 1.12 and 1.23; 95% confidence interval: 1.02–1.22 and 1.10–1.38; *p* = 0.01 and *p* < 0.001, respectively). Conclusions: The 3D-AD quantification derived from deformable registration of serial CT images serves as a marker of lung architectural distortion and a prognostic predictor in patients with IPF.

## 1. Introduction

Idiopathic pulmonary fibrosis (IPF) is a chronic, fibrosing interstitial lung disease (ILD) of unknown cause that is associated with radiological and histologic features of usual interstitial pneumonia (UIP). IPF is characterized by progressive worsening of dyspnea and lung function, with a poor prognosis [1,2]. Recently, antifibrotic drugs have been used in the treatment of patients with IPF [1]. Therefore, monitoring of disease progression has become crucial in the management of patients with IPF.

IPF progression usually manifests as an exacerbated UIP pattern, i.e., increased size and number of honeycomb cysts, and greater degree of traction bronchiectasis and bronchiolectasis [3,4]. Computer-based measurements of the lesion extent on computed tomography (CT) images can provide more objective measures for quantitatively evaluating the disease extent; simultaneously, these methods can reveal decreasing lung volume with disease progression [5,6]. The lung volume reduction with morphological distortion is due to alveolar collapse with UIP-pattern fibrosis [7]. However, slight changes in the lesion volume might be overlooked or underestimated during the initial stage of IPF, even by the traditional computer-based technologies [8]. Therefore, objective evaluation methods for lung distortion are imperative to identify the early stage of progression of UIP-pattern fibrosis. Architectural distortion reflects abnormal displacement of pulmonary structures, such as vessels and airways, results in a distorted appearance of lung anatomy, and is an important CT feature of lung fibrosis progression [9,10]. Deformable image registration (DIR) is widely used for overlapping multiple images with deformation [11,12]. In performing DIR of baseline and follow-up CT images, the three-dimensional displacement (3D-AD) of the lung structures can be quantified. We hypothesized that 3D-AD may be related to pulmonary shrinkage due to lung fibrosis at the corresponding sites and other partial compensatory expansions and have the potential to assess local lung architectural distortion.

Thus, the purpose of this study was to investigate whether 3D-AD can serve as a quantitative parameter for the chronological progression of lung fibrosis and as a prognostic predictor in patients with IPF. 

## 2. Materials and Methods

### 2.1. Study Design and Patient Population

The Institutional Review Board of Kanagawa Cardiovascular and Respiratory Center approved this retrospective, single-center study and waived the requirement for obtaining informed patient consent (KCRC-20-0066). All consecutive patients with ILD who visited our center for the first time from January 2016 to March 2017 were enrolled (Figure 1). Among 588 patients, we excluded patients who had previously undergone thoracic surgery (*n* = 25), severe tuberculosis or other infectious disease (*n* = 26), lung cancer (*n* = 12), pneumothorax (*n* = 5), heart failure (*n* = 1), as well as those who had not undergone CT performed in our center or follow-up CT at 9 to 15 months after initial CT (*n* = 188). Eight patients were excluded because of unavailable image registration for motion artifacts. We also excluded a total of 208 ILD patients with other causes of UIP pattern, i.e., domestic and occupational environmental exposure (*n* = 15), connective tissue disease (*n* = 117), drug toxicity (*n* = 13), and other idiopathic interstitial pneumonias (*n* = 63), such as cryptogenic organizing pneumonia and interstitial pneumonia with autoimmune features [13,14]. The final study population comprised 114 patients. Clinical characteristics, including age, sex, and smoking history, were collected from medical records. Baseline pulmonary function test (PFT) results were acquired within 3 months of the initial CT scans.

### 2.2. CT Data Acquisition

All baseline and follow-up CT images were obtained using a 320-row multidetector CT (MDCT; Aquilion ONE GENESIS, Canon Medical Systems, Otawara, Japan) at full inspiration with 320 rows × 0.5 mm collimation and a tube voltage of 120 kVp with automatic tube current modulation. The reconstruction slice thickness was set at 0.5 mm in 0.5 mm increments. Images were reconstructed using a standard algorithm of filtered back-projection with the FC03 reconstruction kernel, and the field of view was 32 cm (interquartile range (IQR), 32–34 cm). The median effective doses of initial and follow-up CT scans were 5.9 mSv (IQR, 5.0–8.1 mSv) and 5.5 mSv (IQR, 4.9–6.2 mSv), respectively. These data were calculated based on the dose-length product (mGy·cm) and a k-factor of 0.014 (mSv·mGy^−1^·cm^−1^) [15]. 

### 2.3. Quantitative Evaluation

#### 2.3.1. Architectural Distortion Measurement by 3D-AD

Architectural distortion measurement of 3D-AD was performed by the dedicated software on Ziostation2 (QZIP-ILD, ver 0.9, Ziosoft, Inc., Tokyo, Japan; Figure 2). The image registration used mutual information as the image similarity function and estimated the joint probability distribution between the initial and follow-up CT images. Deformations were modeled with cubic B-splines [16]. Following automated lung segmentation, rigid registration of the initial and follow-up CT images was performed. Follow-up CT images were deformably registered to match initial CT images by using landmarks, such as bones, large vessels, bronchial trees, and peripheral pulmonary vessels. After the deformable registration was completed, the three-dimensional displacement of each pixel of the lung field was obtained. Average displacement of the lung-field pixels, i.e., 3D-AD, was calculated. The 3D-AD of the pulmonary structures was also shown as a color map on chest CT images (Figure 2). 

#### 2.3.2. ILD Characterization and Lesion Volume

Total CT lung volume and lesion volumes were measured using quantification software (QZIP-ILD, Ziosoft, Inc.), which is an automatic measurement system that uses a deep-learning-based algorithm [17]. It classified the CT lung field into eight patterns: “normal”, “ground-glass opacity (G)”, “consolidation (C)”, “fibrosis (F)”, which is consolidation with traction bronchiectasis, “reticulation (R)”, “honeycombing (H)”, “traction bronchiectasis (T)”, and “emphysema (E)”. The software was designed to detect fibrosis showing dense air-space consolidation on CT as “consolidation (C)” and “fibrosis (F; consolidation with traction bronchiectasis)”. The volumes of eight-pattern lung categories were computed automatically. A total fibrotic lesion represented the sum of consolidation, fibrosis, reticulation, honeycombing, and traction bronchiectasis. We expressed these lesion-pattern volumes as a percentage of the predicted total lung capacity [8]. Furthermore, we classified the lung field into the subpleural lung region (the outer part of the lung within a 10 mm depth to the visceral pleura) and inner lung region (the other part of the lung). We adopted a 10 mm depth for the classification of subpleural and inner lung regions because a Reid’s secondary lobule is approximately 10 mm in size [18], and the subpleural region is predominantly affected in the IPF patients.

### 2.4. Relationship of Lung Quantification Results and Prognosis

The patients were followed up from January 2016, and the survival status of each patient was confirmed on 31 December 2021. Prognostic information was obtained using electronic medical records. We investigated the presence of death from any cause, including deaths from respiratory failure or lung cancer.

### 2.5. Statistical Analysis

Data were statistically analyzed using SPSS version 27 software (IBM, Armonk, NY, USA). All numeric data are reported as the median with interquartile range. Normality was determined using the Shapiro–Wilk test. Normally distributed values were compared using an unpaired Student’s *t*-test, and non-normally distributed values were compared using the Wilcoxon’s signed-rank test. The optimal cutoff value of 3D-AD for predicting death was the Youden index in the receiver operating characteristics curve. Patients were divided into two groups according to this cutoff value, cumulative survival curves were generated using the Kaplan–Meier method, and statistically significant differences were evaluated using the Log-rank test. Univariate Cox regression analysis was applied to assess the association of clinical characteristics with the time to death or censoring from initial CT imaging. Variables with *p*-values < 0.05 determined by univariate logistic regression analysis were inputted as variables in a multivariate logistic regression analysis. A *p*-value < 0.05 was considered statistically significant.

## 3. Results

### 3.1. Patient Characteristics

The final study population comprised 114 consecutive patients with IPF diagnosed by a multidisciplinary method according to the IPF guidelines [1,2]. Among these patients, 62 (54%) patients had a bronchoalveolar lavage, and 41 (36%) patients had a surgical lung biopsy. There were 84 males and 30 females, with a median age of 69 (IQR, 64–74) years at presentation. The ILD of our IPF population was a UIP, a probable UIP, and an indeterminate for UIP pattern in 23, 21, and 70 of 114 patients, respectively [2]. 

Table 1 summarizes the patient characteristics. Here, 85 (75%) patients were former or current smokers, the median follow-up time was 55 (IQR, 36–62) months, and 35 (31%) patients died of any cause. The median % forced vital capacity (FVC) predicted was 82% (IQR, 70–97%). The % diffusing capacity of the lung for the carbon monoxide (DLco) predicted measurement at baseline was available for 113 (99%) patients. The median %DLco predicted was 77% (IQR, 56–94%). Ten (8%) patients had home oxygen therapy (HOT) at the time of enrollment. Some or all agents were duplicated, and antifibrotic, corticosteroid, and other immunosuppressive agents were administered to 40 (35%), 39 (34%), and 13 (30%) patients, respectively, at some time point.

### 3.2. Quantitative Evaluation

#### 3.2.1. Architectural Distortion Measurement by 3D-AD

Figure 3, Figure 4 and Figure 5 are representative 3D-AD color maps. The displacement of the lung structures was not homogenous in the lung. In some cases, marked differences were seen in the right and left lungs in the 3D-AD color maps (Figure 4). As shown in Figure 3, displacement of the lung structures was larger in the subpleural lung region than that in the inner region (5.2 mm (IQR, 3.6–7.1 mm) vs. 4.7 mm (IQR, 3.0–6.4 mm), respectively, *p* < 0.001). In the subpleural lung region, the fibrotic lesion volume change was also larger than that in the inner lung region (0.70% (IQR, 0.22–1.60%) vs. 0.21% (IQR, 0.004–1.12%), respectively, *p* < 0.001). In some cases, the displacement of peripheral structures was observed before increases in fibrotic lesion volume were obvious on follow-up CT (Figure 5).

#### 3.2.2. ILD Characterization and Lesion Volume

Table 2 and Table 3 summarize the initial CT results. The fibrotic lesion volume and that normalized for predicted total whole-lung capacity were 277 mL (IQR, 143–526 mL) and 5.8% (IQR, 3.1–10.8%), respectively. All patterns of lesions were significantly larger in the subpleural lung region than in the inner region. The fibrotic lesion was significantly larger in the subpleural lung region than in the inner region (4.98% (IQR, 2.73–9.13%) vs. 0.70% (IQR, 0.22–2.02%), respectively, *p* < 0.001). On follow-up CT, the fibrotic lesion volume increased by 52.1 mL (IQR, 7.9–131.2 mL), and its predicted TLC increased by 1.1% (IQR, 0.2–2.7%), whereas the total lung volume decreased by 183 mL (IQR, 131–469 mL) and its predicted TLC by 3.5% (IQR, 2.37–9.54%). Table 4 summarizes the changes in lesion patterns and 3D-AD between initial and follow-up CT images in the whole, subpleural, and inner lung regions.

### 3.3. Relationship of Lung Quantification Results and Prognosis

The survival curves comparing 3D-AD are shown in Figure 6. The optimal cutoff value for 3D-AD was 4.29 mm, with a sensitivity of 97% (95%CI: 85–99), specificity of 51% (95%CI: 39–62), and an AUC of 0.69 (95%CI: 0.59–0.79) for predicting all-cause mortality. Log-rank tests revealed a significant difference in overall survival between the two groups based on the 3D-AD (3D-AD ≥ 4.29 mm and 3D-AD < 4.29 mm, *p* < 0.001). Table 5 shows univariable Cox regression analysis results of the CT-derived lung lesion volume normalized for predicted total lung capacity and 3D-AD, and their associations with all-cause mortality. Univariable Cox regression analysis showed that a large change in fibrotic lesion volume in the whole lung and in the subpleural lung region were significantly associated with mortality (hazard ratio, 1.18; 95% confidence interval (CI), 1.07–1.30; *p* = 0.001, and hazard ratio, 1.85; 95% CI, 1.55–2.22; *p* < 0.001, respectively). The change in the fibrotic lesion volume in the inner lung region was not significantly associated with mortality (hazard ratio, 0.96; 95% CI, 0.83–1.10; *p* = 0.18). 

Univariable Cox regression analysis showed that 3D-AD in the subpleural lung region was associated with mortality (hazard ratio, 1.20; 95% CI, 1.10–1.30; *p* < 0.001). We performed multivariate logistic analysis of significant CT results and clinical factors. Multivariate logistic analysis, including age, sex, FVC%, 3D-AD in the subpleural lung region, and change in fibrotic lesion volume of the whole lung normalized for predicted total lung capacity, showed FVC% (hazard ratio, 0.95; 95% CI, 0.94–0.97; *p* < 0.001), 3D-AD in the subpleural region (hazard ratio, 1.12; 95% CI, 1.02–1.22; *p* = 0.01), and change in the fibrotic lesion volume of the whole lung (hazard ratio, 1.23; 95% CI, 1.10–1.38; *p* < 0.001) were independent predictors for mortality. Age (*p* = 0.57) and sex (*p* = 0.95) were not significant predictors. Multivariate logistic analysis, including the presence of baseline HOT, HOT, and 3D-AD in the subpleural region were independent predictors for mortality (hazard ratio, 2.58; 95% CI, 1.01–6.12; *p* < 0.03, and 1.19; 95% CI, 1.01–1.29; *p* < 0.001, respectively).

## 4. Discussion

In this study, we quantitatively evaluated architectural distortion of the lung in the serial CT images of patients with IPF using DIR. The median 3D-AD based on DIR between the initial and follow-up CT images was significantly larger in the subpleural lung region than in the inner region. Our volumetric analysis of the lung lesions showed that the fibrotic lesion volume on baseline CT and the annual increase in fibrotic lesion volume in the subpleural region were significantly larger than those in the inner region, accompanied by a decrease in total lung volume. Though the majority (61%) of patients with IPF presented an HRCT pattern of indeterminate for UIP in this study, these results were comparable with those of previous studies showing the subpleural predominance in UIP-pattern fibrosis [1,19,20]. Therefore, we think that 3D-AD in the subpleural region would correspond to the architectural distortion due to progression of lung fibrosis. Multivariate logistic analysis showed that a larger 3D-AD in the subpleural region was one of the independent predictors for mortality, alongside a decreased %FVC and increased fibrotic lesion volume of the whole lung. It is well known that a decreased %FVC and increased extent of fibrotic lesion on CT are associated with poor prognosis [21,22,23]. We propose 3D-AD as a useful measure to evaluate disease progression and prognosis in patients with IPF.

Architectural distortion is an abnormal appearance of secondary pulmonary lobule shape or size with evidence of volume loss [24]; however, it has not been fully evaluated quantitatively. Chassagnon et al. introduced elastic registration to evaluate lung deformation [25]. Sun et al. obtained deformation maps of the lungs by performing elastic registration of baseline and follow-up CT images of patients with IPF [26]. These authors demonstrated that the logarithm of the Jacobian (log_jac) determinant for each voxel of the deformation field was correlated with pulmonary function test outcomes in patients with IPF. However, they did not compare the Jacobian (log_jac) determinant and patient survival. To the best of our knowledge, our study is the first to quantitatively evaluate architectural distortion by applying the DIR technique (3D-AD) and compare it with the prognosis of patients with IPF. Further studies to compare the 3D-AD and the Jacobian (log_jac) determinant should be performed.

We observed heterogenous displacement of lung structures in the 3D-AD color map; for example, differences between the right and left lungs, as well as the subpleural and inner regions of the lung, were evident. We observed that the increases in the fibrotic lesion volume and 3D-AD were larger in the subpleural lung region compared to those in the inner region. These results support our hypothesis that 3D-AD is related to architectural distortion due to fibrosis. In the current study, we observed a larger amount of fibrosis on initial CT and an increase in fibrosis on follow-up CT in the subpleural region. This might be explained by the self-perpetuating fibrosis, which is a major pathological pathway of UIP fibrosis, i.e., fibroblast activation and differentiation into myofibroblasts occur for any cause, and once established, structural tissue changes and the profibrotic milieu form a feed-forward loop, leading to self-perpetuating fibrosis [27]. 

The recent guideline indicates radiological evidence of disease progression in UIP pattern fibrosis as an increased extent or severity of traction bronchiectasis and bronchiolectasis, new ground-glass opacity with traction bronchiectasis, new fine reticulation, increased extent or increased coarseness of reticular abnormality, new or increased honeycombing, and increased lobar volume [1]. An increase in lesion volume on CT and a decrease in lung volume can be objectively measured using computer-based software. The change in lesion extent, however, is not large in UIP-pattern fibrosis; for example, a previous report showed the annual change in fibrotic lesion volume was 2% of the predicted total lung capacity of patients with IPF without antifibrotic drug use, and 0.14% in those using antifibrotic drugs [28]. This may be because UIP-pattern fibrosis is accompanied by alveolar collapse (volume decrease) [29], and in smokers, emphysematous lung (volume increase), and these simultaneous phenomena will mask volume loss, leading to underestimation of the progression by the lesion volumetric analysis. We stress that the 3D-AD color map could demonstrate local parenchymal displacement in patients with a small amount of increase in the fibrotic lesion extent, as shown in Figure 5. We believe the 3D-AD color map could help raise awareness of fibrosis progression in such mild cases. In the recent clinical practice guideline [1], progressive pulmonary fibrosis (PPF) is newly defined as at least two of three criteria (worsening symptoms, radiological progression, and physiological progression) occurring within the past year with no alternative explanation in patients with an ILD other than IPF. In addition to computer-based quantitative assessments of the fibrotic lesion extent, 3D-AD of annual HRCT could be useful to promptly detect radiological progression of PPF and possibly influence prognosis. 

This study has several limitations. First, this was a retrospective study of a single center. A multi-center study involving a larger population is needed to evaluate the usefulness of 3D-AD. Second, the degree of inspiration of patients cannot be well controlled, which might have impacted our results. However, patients with IPF were well trained in respiratory performance evaluation because they underwent repeated follow-up evaluations, including PFTs, which probably minimized the variability. Heterogenous distribution of 3D-AD cannot be explained by simple differences in the inspiratory level. Third, 3D-AD is an absolute value, not a vector. We cannot decipher whether the lung is shrinking or stretching from 3D-AD. We observed an increased fibrotic lesion volume and decreased total lung volume on follow-up CT, and we speculated that 3D-AD would show mixed values of the lung shrinkage and local stretching due to patchy UIP-pattern fibrosis. The development of methods that can indicate the direction of displacement and distinguish local shrinking from local stretching of the lung parenchyma may be required. Fourth, we demonstrated the potential of the 3D-AD color-map images, but the effects on the clinical practice were not thoroughly investigated. A comparison study of the clinical course of IPF patients and qualitative evaluation of the color map should be conducted.

## 5. Conclusions

In conclusion, we measured 3D-AD using deformable registration of serial CT images in patients with IPF, as a quantitative marker indicating architectural distortion. We found larger 3D-AD and a greater increase in the fibrotic lesion volume in the subpleural region compared to those in the inner region of the lung. Furthermore, larger 3D-AD in the subpleural region was independently associated with a poor prognosis in patients with IPF. The 3D-AD may be a useful marker for evaluating fibrosis progression, potentially aiding in establishment of appropriate management strategies for patients with IPF.

## Figures and Tables

**Figure 1 diagnostics-14-01650-f001:**
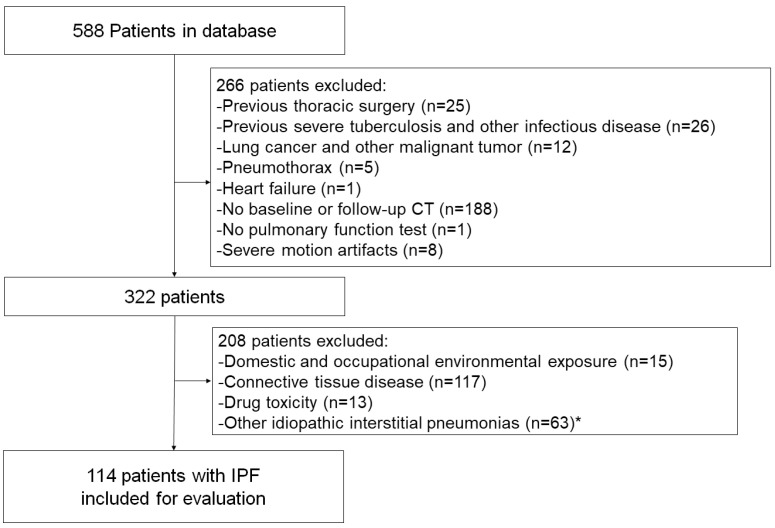
Flowchart of the study population selection and exclusion criteria. IPF, idiopathic pulmonary fibrosis. * Other idiopathic interstitial pneumonias include cryptogenic organizing pneumonias and interstitial pneumonia with autoimmune features.

**Figure 2 diagnostics-14-01650-f002:**
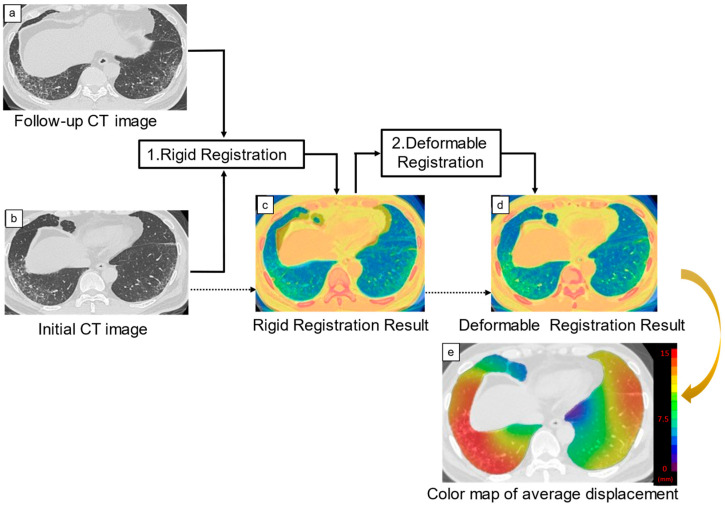
Creation of the three-dimensional average displacement (3D-AD) color map. Following the rigid registration of the initial and follow-up computed tomography (CT) images, follow-up CT images were deformably registered to match initial CT images with extracted landmarks, such as bones, large vessels, bronchial trees, and peripheral pulmonary vessels. Deformable registration was performed to overlap these paired landmarks completely. The 3D-AD of each pixel was calculated and shown as a 3D-AD color map. The color bar indicates the range of displacement.

**Figure 3 diagnostics-14-01650-f003:**
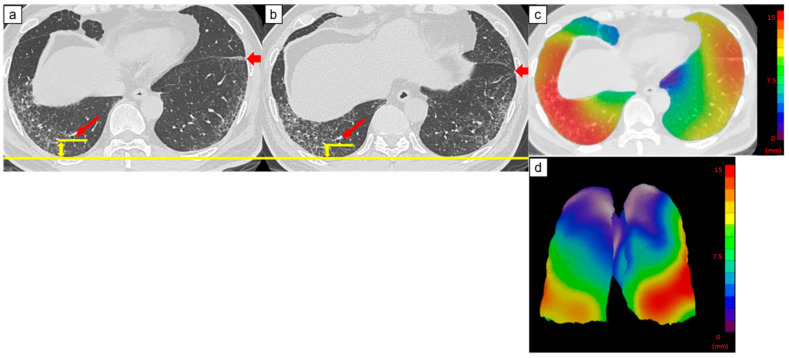
Computed tomography (CT) images of a 69-year-old man with idiopathic pulmonary fibrosis and three-dimensional average displacement (3D-AD) color maps. (**a**) Initial CT image of the lung base showed mild subpleural reticulation. (**b**) Follow-up CT image of the lung base after one year shows increased subpleural reticulation. The fissure deviated backward, indicating the decreased lower lobe volume (thick arrows). The distance between the peripheral bronchi (arrows) and pleura also decreased (yellow lines). (**c**,**d**) Color map of the 3D-AD demonstrates greater deviation (red in the color map), corresponding to the subpleural reticulation in the original CT images. In this case, 3D-AD in the subpleural/inner region = 7.4 mm/7.2 mm.

**Figure 4 diagnostics-14-01650-f004:**
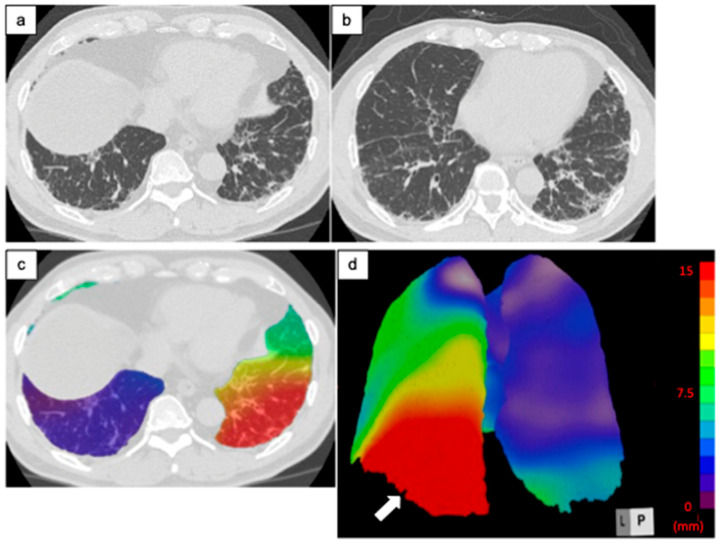
Computed tomography (CT) images of a 68-year-old man with idiopathic pulmonary fibrosis and three-dimensional average displacement (3D-AD) color maps. (**a**) Initial CT image shows subpleural reticulation in the lung base. (**b**) Follow-up CT image after one year shows an increase in subpleural reticulations predominantly in the posterior areas of the left lung base. (**c**,**d**) Color map of the 3D-AD shows a greater deviation (red in the color map) in the posterior areas of the left lung base (white arrow), corresponding to the original CT images. In this case, 3D-AD in the whole lung/lower lobe of the left lung = 5.9 mm/14 mm.

**Figure 5 diagnostics-14-01650-f005:**
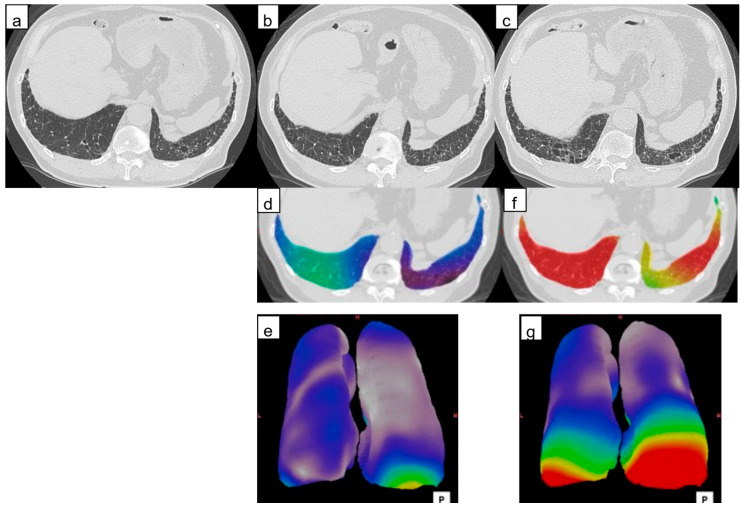
Computed tomography (CT) images of a 77-year-old man with idiopathic pulmonary fibrosis and three-dimensional average displacement (3D-AD) color maps. (**a**) Initial CT image of the lung base showed subpleural interstitial abnormality. (**b**) Follow-up CT image after one year shows a little increase of subpleural lesions in the posterior areas of the right lung base. (**c**) Follow-up CT image after two years shows obvious progression of fibrosis at the lung base. (**d,e**) The 3D-AD color map shows yellow scale in the posterior areas of the right lung base. (**f**,**g**) Color map of the 3D-AD shows a greater deviation (red in the color map) observed. Local displacement of peripheral structures is evident before lung abnormalities and lung volume loss are clearly visible at the follow-up CT.

**Figure 6 diagnostics-14-01650-f006:**
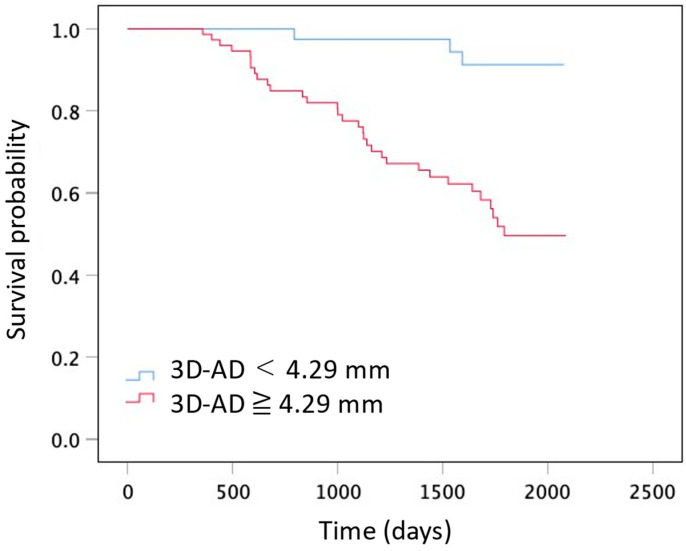
Graphs show Kaplan–Meier distribution of survival time. Comparison between two groups on the basis of 3D-AD (3D-AD ≥ 4.29 mm and 3D-AD < 4.29 mm; Log-rank *p* < 0.001).

**Table 1 diagnostics-14-01650-t001:** Patient clinical characteristics.

	All Patients (*n* = 114)	Hazard Ratio (95% CI)	*p*-Value
Age (y) ^a^	69 (64–74)	1.04 (0.99–1.08)	0.13
Sex (M/F)	84 (74)/30 (26)	0.75 (0.37–1.50)	0.44
History of smoking	85 (75)	0.78 (0.38–1.63)	0.51
Smoking index (no. of pack-years) ^a^	20 (0–46)	1.00 (0.99–1.01)	0.96
Baseline home oxygen therapy (no. of patients)	10 (8)	3.31 (1.44–7.61)	0.005
Baseline pulmonary function test results			
FVC (% predicted) ^a^	82 (70–97)	0.96 (0.94–0.97)	<0.001
DLco (% predicted) ^a,b^	77 (56–94)	0.97 (0.96–0.98)	<0.001
Bronchoalveolar lavage (no. of patients)	62 (54)		
Alveolar macrophages (%) ^a^	71 (60–83)	0.99 (0.97–1.01)	0.37
Alveolar lymphocyte (%) ^a^	17 (8–26)	0.98 (0.94–1.01)	0.18

Unless otherwise specified, data are numbers of patients, with percentages in parentheses. ^a^ Numbers in parentheses are the interquartile range. ^b^ Missing data for one patient. FVC, forced vital capacity; DLco, diffusion capacity of the lung for carbon monoxide; CI, confidence intervals.

**Table 2 diagnostics-14-01650-t002:** Medians and interquartile ranges of initial CT lesion patterns.

Lesion Patterns ^b^	Whole Lung	Subpleural Region ^d^	Inner Region	*p*-Value ^a^
Emphysema (%)	0.59 (0.14–1.67)	0.44 (0.11–1.12)	0.09 (0.01–0.35)	<0.001
Consolidation + Fibrosis (%)	1.59 (1.00–2.72)	1.14 (0.58–1.75)	0.19 (0.05–0.52)	<0.001
Ground-glass opacity (%)	5.38 (3.40–8.32)	3.61 (2.27–4.73)	1.55 (0.64–2.66)	<0.001
Reticulation (%)	0.52 (0.24–1.42)	2.81 (1.50–5.36)	0.20 (0.06–0.83)	<0.001
Honeycombing + Traction bronchiectasis (%)	3.48 (1.84–6.44)	0.31 (0.09–0.91)	0.18 (0.08–0.44)	<0.001
Normal (%)	58.1 (40.1–72.4)	24.7 (16.7–31.8)	32.4 (23.6–41.2)	<0.001
Fibrotic lesion (%) ^c^	5.84 (3.06–10.8)	4.98 (2.73–9.13)	0.70 (0.22–2.02)	<0.001

Numbers in parentheses are the interquartile range. ^a^ Wilcoxon’s signed-rank test to compare each lesion pattern between the subpleural and inner lung region. ^b^ CT lung lesion volume normalized for predicted total lung capacity. ^c^ Sum of values of consolidation, fibrosis, reticulation, honeycombing, and traction bronchiectasis. ^d^ The 10 mm-width area of the lung surface.

**Table 3 diagnostics-14-01650-t003:** Initial CT lesion patterns and their associations with mortality according to univariable Cox regression analysis.

	Whole Lung		Subpleural Region ^c^		Inner Region	
Lesion Patterns ^a^	Hazard Ratio (95% CI)	*p*-Value	Hazard Ratio (95% CI)	*p*-Value	Hazard Ratio (95% CI)	*p*-Value
Emphysema (%)	1.05 (0.96–1.14)	0.32	1.09 (0.93–1.29)	0.30	1.08 (0.90–1.30)	0.39
Consolidation + Fibrosis (%)	1.6 (1.40–1.83)	<0.001	2.22 (1.75–2.82)	<0.001	2.30 (1.65–3.22)	<0.001
Ground-glass opacity (%)	1.12 (1.06–1.18)	<0.001	1.17 (1.04–1.32)	0.008	1.23 (1.12–1.34)	<0.001
Reticulation (%)	1.17 (1.07–1.27)	0.001	1.22 (1.12–1.32)	<0.001	1.55 (1.26–1.90)	<0.001
Honeycombing + Traction bronchiectasis (%)	1.16 (1.09–1.23)	<0.001	1.24 (1.08–1.42)	0.002	1.63 (1.28–2.08)	<0.001
Normal (%)	0.67 (0.35–1.27)	0.218	0.89 (0.86–0.93)	<0.001	0.92 (0.90–0.95)	<0.001
Fibrotic lesion (%) ^b^	1.10 (1.06–1.14)	<0.001	1.17 (1.10–1.23)	<0.001	1.26 (1.14–1.38)	<0.001

Data in parentheses are the 95% confidence intervals. ^a^ CT lung lesion volume normalized for predicted total lung capacity. ^b^ Sum of values of consolidation, fibrosis, reticulation, honeycombing, and traction bronchiectasis. ^c^ The 10 mm-width area of the lung surface. CI, confidence intervals.

**Table 4 diagnostics-14-01650-t004:** Median and interquartile range values of changes in CT lung lesion volume normalized for the predicted total lung capacity and 3D-AD.

	Whole Lung	Subpleural Region ^c^	Inner Region	*p*-Value ^a^
Categorical change in CT lesion variables ^b^				
ΔEmphysema (%)	0.22 (0.03–0.90)	0.10 (0.01–0.34)	0.05 (−0.00–0.47)	0.40
ΔConsolidation + Fibrosis (%)	0.25 (0.56–0.71)	0.15 (0.02–0.38)	0.06 (0.00–0.31)	0.01
ΔGround-glass opacity (%)	0.12 (−1.37–1.43)	−0.06 (−0.75–0.58)	0.10 (−0.54–0.87)	0.001
ΔReticulation (%)	0.52 (−0.07–1.35)	0.36 (−0.05–0.87)	0.04 (−0.04–0.45)	0.003
ΔHoneycombing + Traction bronchiectasis (%)	0.16 (0.01–0.81)	0.08 (0.01–0.34)	0.047 (−0.01–0.26)	<0.001
ΔNormal (%)	0.00 (−0.01–0.03)	−1.82 (−4.01–−0.06)	−2.54 (−8.47–0.97)	0.02
ΔFibrotic lesion (%)	1.08 (0.15–2.69)	0.70 (0.22–1.60)	0.21 (0.00–1.12)	<0.001
ΔTotal CT lung volume (cc)	−183 (−469–131)			
3D-AD (mm)	5.0 (3.3–6.7)	5.2 (3.6–7.1)	4.7 (3.0–6.4)	<0.001

Numbers in parentheses are the interquartile range. ^a^ Wilcoxon’s signed-rank test to compare CT parameters between subpleural and inner lung regions. ^b^ Result of subtraction initial CT lung lesion from follow up CT lung lesion. ^c^ The 10 mm-width area of the lung surface. 3D-AD, three-dimensional average displacement.

**Table 5 diagnostics-14-01650-t005:** CT lung lesion volume normalized for the predicted total lung capacity and 3D-AD, and their associations with mortality according to univariable Cox regression analysis.

	Whole Lung		Subpleural Region ^b^		Inner Region	
	Hazard Ratio (95% CI)	*p*-Value	Hazard Ratio (95% CI)	*p*-Value	Hazard Ratio (95% CI)	*p*-Value
Categorical change in CT lesion variables ^a^						
ΔEmphysema (%)	1.12 (0.97–1.30)	0.134	1.78 (1.30–2.45)	<0.001	1.02 (0.82–1.26)	0.86
ΔConsolidation + Fibrosis (%)	1.41 (0.94–2.11)	0.093	2.33 (1.42–3.81)	0.001	0.80 (0.44–1.43)	0.45
ΔGround-glass opacity (%)	0.93 (0.87–1.00)	0.05	0.94 (0.76–1.16)	0.054	0.89 (0.80–0.99)	0.03
ΔReticulation (%)	1.20 (1.01–1.42)	0.035	1.64 (1.26–2.13)	<0.001	0.95 (0.69–1.30)	0.73
ΔHoneycombing + Traction bronchiectasis (%)	1.65 (1.28–2.13)	<0.001	2.00 (1.55–2.58)	<0.001	0.93 (0.61–1.40)	0.73
ΔNormal (%)	1.02 (0.78–1.34)	0.889	0.76 (0.69–0.83)	<0.001	1.01 (0.72–1.43)	0.96
ΔFibrotic lesion (%)	1.18 (1.07–1.30)	0.001	1.85 (1.55–2.22)	<0.001	0.96 (0.83–1.10)	0.18
3D-AD (mm)	1.20 (1.20–1.30)	<0.001	1.20 (1.10–1.30)	<0.001	1.20 (1.10–1.30)	<0.001

Data in parentheses are the 95% confidence intervals. ^a^ Result of subtracting the initial CT lung lesion from the follow-up CT lung lesion. ^b^ The 10 mm-width area of the lung surface. 3D-AD, three-dimensional average displacement; CI, confidence intervals.

## Data Availability

The data will be shared upon reasonable request to the corresponding author.

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
