# Peer review of "Evaluation of Progressive Architectural Distortion in Idiopathic Pulmonary Fibrosis Using Deformable Registration of Sequential CT Images"

_diagnostics, 2024, doi:10.3390/diagnostics14151650_

Round 1

Reviewer 1 Report

Comments and Suggestions for Authors

Dear authors,

congratulations on your article. I find the following context very interesting and relevant. Here are my considerations.

-Please note that the background of the abstract contains the aim of the study. Correct it to make it consistent with the paragraph.

-The introduction is well-structured, and the objectives of the study are clearly stated.

-In line 81, the number 189 is not consistent with Figure 1. Please correct it.

-The quality of Figure 1 could be improved.

-The methods are well articulated and well supported by the bibliography.

-The results are extensively written and validated by good statistical analysis.

-The results are supported by valid tables.

-The overall quality of the figures should be improved, considering that the quality of the figures in the supplement is good.

-The discussion clearly states the novelty introduced by this study, namely testing an innovative tool to assess disease progression and evaluate prognostic valuation in patients with IPF. The discussion continues with a thorough evaluation of the results obtained, comparing them with the current research already published. The limitations are well highlighted.

-The conclusions are consistent with the study objectives.

-It would be interesting to elaborate on the present work by knowing whether the enrolled patients had oxygen therapy or ventilatory support at the time of enrollment or follow-up. This is especially important to associate this data with the outcome values.

-The bibliography is valid.

Author Response

Reviewer 1

Dear authors,

congratulations on your article. I find the following context very interesting and relevant. Here are my considerations.

We wish to express our appreciation to the reviewer for the insightful comments on our paper. The comments have helped us significantly improve the paper.

-Please note that the background of the abstract contains the aim of the study. Correct it to make it consistent with the paragraph.

We thank the reviewer for this comment. In accordance with the reviewer’s comment, we have changed background of the abstract to “Monitoring the progression of idiopathic pulmonary fibrosis (IPF) using CT primarily focuses on assessing the extent of fibrotic lesions, without considering the distortion of lung architecture.”

-The introduction is well-structured, and the objectives of the study are clearly stated.

We thank the reviewer for the positive comment.

-In line 81, the number 189 is not consistent with Figure 1. Please correct it.

We thank the reviewer for the careful review. As suggested, we have corrected the number to 188.

-The quality of Figure 1 could be improved.

We thank the reviewer for this comment. In accordance with reviewer’s comment, we have replaced Figure 1.

-The methods are well articulated and well supported by the bibliography.

We thank the reviewer for the positive comment.

-The results are extensively written and validated by good statistical analysis.

We thank the reviewer for the positive comment.

-The results are supported by valid tables.

We thank the reviewer for the positive comment.

-The overall quality of the figures should be improved, considering that the quality of the figures in the supplement is good.

We thank the reviewer for this comment. In accordance with reviewer’s comment, we have improved the quality of all figures.

-The discussion clearly states the novelty introduced by this study, namely testing an innovative tool to assess disease progression and evaluate prognostic valuation in patients with IPF. The discussion continues with a thorough evaluation of the results obtained, comparing them with the current research already published. The limitations are well highlighted.

We thank the reviewer for the positive comment.

-The conclusions are consistent with the study objectives.

We thank the reviewer for the positive comment.

-It would be interesting to elaborate on the present work by knowing whether the enrolled patients had oxygen therapy or ventilatory support at the time of enrollment or follow-up. This is especially important to associate this data with the outcome values.

We thank the reviewer for this comment. Ten (8%) patients had oxygen therapy at the time of enrollment. In accordance with the reviewer’s comment, we have added text to the Result in line 191-192.

-The bibliography is valid.

We thank the reviewer for the positive comment.

Reviewer 2 Report

Comments and Suggestions for Authors

Many thanks for the opportunity to review the manuscript entitled: Evaluation of progressive architectural distortion in idiopathic pulmonary fibrosis using deformable registration of sequential CT images. However, in the method authors should specify that diagnosis of IPF following the current international guidelines.  Therefore it isn’t specified the exclusion of all the other causes of UIP pattern as all of these disease: domestic and occupational environmental exposures, connective tissue disease, and drug toxicity. Therefore, all these conditions should be included in the exclusion criteria.

Raghu, G., Collard, H. R., Egan, J. J., Martinez, F. J., Behr, J., Brown, K. K., ... & Schunemann, H. J. (2011). An official ATS/ERS/JRS/ALAT statement: idiopathic pulmonary fibrosis: evidence-based guidelines for diagnosis and management. American journal of respiratory and critical care medicine183(6), 788-824.

In how many patients were performed the BAL?

On HRCT did authors consider all true UIP pattern in IPF? Or did they include also a probable or indeterminate UIP pattern? This issue should be specified

Author Response

Reviewer 2

Many thanks for the opportunity to review the manuscript entitled: Evaluation of progressive architectural distortion in idiopathic pulmonary fibrosis using deformable registration of sequential CT images. However, in the method authors should specify that diagnosis of IPF following the current international guidelines. Therefore it isn’t specified the exclusion of all the other causes of UIP pattern as all of these disease: domestic and occupational environmental exposures, connective tissue disease, and drug toxicity. Therefore, all these conditions should be included in the exclusion criteria.

Raghu, G., Collard, H. R., Egan, J. J., Martinez, F. J., Behr, J., Brown, K. K., ... & Schunemann, H. J. (2011). An official ATS/ERS/JRS/ALAT statement: idiopathic pulmonary fibrosis: evidence-based guidelines for diagnosis and management. American journal of respiratory and critical care medicine, 183(6), 788-824.

We wish to express our appreciation to the reviewer for the insightful comments on our paper. The comments have helped us significantly improve the paper. There were 15 patients with domestic and occupational environmental exposure, 117 patients with connective tissue disease, 13 patients with drug toxicity, 63 patients with other idiopathic interstitial pneumonias. We have revised the Figure 1 and the text in the Materials and Methods as follows:

“All consecutive patients with ILD who visited our center for the first time from January 2016 to March 2017 were enrolled (Figure 1). Among 588 patients, we excluded patients who had previously undergone thoracic surgery (n = 25), severe tuberculosis or other infectious disease (n = 26), lung cancer (n = 12), pneumothorax (n = 5), heart failure (n = 1), as well as those who had not undergone CT performed in our center or follow-up CT at 9 to 15 months after initial CT (n = 188). Eight patients were excluded because of unavailable image registration for motion artifacts. We also excluded a total of 208 ILD patients with other causes of UIP pattern, i.e., domestic and occupational environmental exposure (n = 15), connective tissue disease (n = 117), drug toxicity (n = 13) and other idiopathic interstitial pneumonias (n = 63), such as cryptogenic organizing pneumonia, interstitial pneumonia with autoimmune features [13]. The final study population comprised 114 patients. Clinical characteristics, including age, sex, and smoking history, were collected from medical records. Baseline pulmonary function test (PFT) results were acquired within 3 months of the initial CT scans.”

In how many patients were performed the BAL?

We thank the reviewer for this comment. There were 62 patients with bronchoalveolar lavage (BAL). Accordingly, we have added text to Results in line 181-182 and cell differential of BAL to Table 1.

On HRCT did authors consider all true UIP pattern in IPF? Or did they include also a probable or indeterminate UIP pattern? This issue should be specified.

We thank the reviewer for the important comment. Our population consisted of 114 IPF patients confirmed by our multidisciplinary discussion and 90% of them underwent BAL or biopsy, but not all patients exhibited UIP pattern. As suggested, we have stated the information as follows in line 184-185:

“The ILD of our IPF population was a UIP, a probable UIP, and an indeterminate UIP pattern in 23, 21, and 70 of 114 patients, respectively. “

Round 2

Reviewer 2 Report

Comments and Suggestions for Authors

The authors have made the requested adjustment.  If authors can add a small description of the CT presentations of idiopathic pulmonary fibrosis (pattern UIP, PROBABLE, INDETERMINATE) in the discussion to make this manuscript easier to read for all physicians. 

Author Response

Reviewer comment

The authors have made the requested adjustment. If authors can add a small description of the CT presentations of idiopathic pulmonary fibrosis (pattern UIP, PROBABLE, INDETERMINATE) in the discussion to make this manuscript easier to read for all physicians.

We wish to express our appreciation to the reviewer for the insightful comments on our paper. The comments have helped us significantly improve the paper. We have revised the text in the Discussion as follows:

“Though the majority (61%) of patients with IPF were presenting HRCT pattern of indeterminate for UIP in this study, these results were comparable with those of previous studies showing the subpleural predominance in UIP pattern fibrosis [1,19,20].”